# Transcriptomic Revelation of Phenolic Compounds Involved in Aluminum Toxicity Responses in Roots of *Cunninghamia lanceolata* (Lamb.) Hook

**DOI:** 10.3390/genes10110835

**Published:** 2019-10-23

**Authors:** Zhihui Ma, Sizu Lin

**Affiliations:** 1Institute for Forest Resources and Environment of Guizhou, Guizhou University, Guiyang 550025, China; zhihuima@fafu.edu.cn; 2State Forestry Administration Engineering Research Center of Chinese Fir, Fuzhou 350002, China; 3College of Forestry, Fujian Agricultural and Forestry University, Fuzhou 350002, China

**Keywords:** Chinese fir, Al toxicity, Acid soil, Flavonoids pathway, Phenylpropanoids metabolism, Phenolic compounds

## Abstract

Chinese fir (*Cunninghamia lanceolata* (Lamb.) Hook.) is one of the most important coniferous evergreen tree species in South China due to its desirable attributes of fast growth and production of strong and hardy wood. However, the yield of Chinese fir is often inhibited by aluminum (Al) toxicity in acidic soils of South China. Understanding the molecular mechanisms of Chinese fir root responses to Al toxicity might help to further increase its productivity. Here we used the Illumina Hiseq4000 platform to carry out transcriptome analysis of Chinese fir roots subjected to Al toxicity conditions. A total of 88.88 Gb of clean data was generated from 12 samples and assembled into 105,732 distinct unigenes. The average length and N50 length of these unigenes were 839 bp and 1411 bp, respectively. Among them, 58362 unigenes were annotated through searches of five public databases (Nr: NCBI non-redundant protein sequences, Swiss-Prot: A manually annotated and reviewed protein sequence database, GO: Gene Ontology, KOG/COG: Clusters of Orthologous Groups of proteins, and KEGG: the Kyoto Encyclopedia of Genes and Genomes database), which led to association of unigenes with 44 GO terms. Plus, 1615 transcription factors (TFs) were functionally classified. Then, differentially expressed genes (DEGs, |log_2_(fold change)| ≥ 1 and FDR ≤ 0.05) were identified in comparisons labelled TC1 (CK-72 h/CK-1 h) and TC2 (Al-72 h/Al-1 h). A large number of TC2 DEGs group were identified, with most being down-regulated under Al stress, while TC1 DEGs were primarily up-regulated. Combining GO, KEGG, and MapMan pathway analysis indicated that many DEGs are involved in primary metabolism, including cell wall metabolism and lipid metabolism, while other DEGs are associated with signaling pathways and secondary metabolism, including flavonoids and phenylpropanoids metabolism. Furthermore, TFs identified in TC1 and TC2 DEGs represented 21 and 40 transcription factor families, respectively. Among them, expression of bHLH, C2H2, ERF, bZIP, GRAS, and MYB TFs changed considerably under Al stress, which suggests that these TFs might play crucial roles in Chinese fir root responses to Al toxicity. These differentially expressed TFs might act in concert with flavonoid and phenylpropanoid pathway genes in fulfilling of key roles in Chinese fir roots responding to Al toxicity.

## 1. Introduction

Chinese fir (*Cunninghamia lanceolata* (Lamb.) Hook.) is an important coniferous evergreen tree species that is widely cultivated in South China due to its desirable attributes of fast growth and production of strong and hardy wood. Based on planting area data in China’s ninth national forest inventory, Chinese fir is the second most dominant tree species in China, with cultivation occurring on approximately 11 million ha of plantations that account for about 6% of all forested land and 14.2% of total plantation area in China. This tree species, which has been planted in southeastern Asia for more than 1000 years [1], is primarily distributed in South China [2]. Generally speaking, soils in south China are largely acidic or strongly acidic, with pH values below 6.0 widespread, and in below 5.5 reported at times from strongly acidic soils.

Globally, 50% of the world’s potential arable land is composed of acid soils [3], which are typically associated with deficiencies of phosphorus and other nutrients, as well as, toxic concentrations of certain metals, including iron (Fe) and Aluminum (Al) [4]. As a major constituent of mineral soils, Al is present in a wide range of primary and secondary minerals [5]. In acid soils Al solubility increases to the point where Al toxicity becomes one of the most important factors limiting plant growth [4,6,7]. Several recent reports have outlined this process in soil where, as pH falls below 5.0, Al is solubilized into phytotoxic Al^3+^ ions from non-toxic Al silicates and oxides [4,8,9].

The most evident symptom of Al toxicity in plants is rapid inhibition of root elongation through destruction of the root apex, which results in insufficient uptake of water and nutrients [10,11]. The root apex, particularly the transition zone, has also been identified as critical for sensing Al^3+^ toxicity and directing responses imparting plant tolerance to otherwise toxic levels of Al^3+^ [12]. In one notable study, Rengel observed that Al toxicity effects on shoots, such as growth inhibition and nutrient deficiency symptoms in aboveground tissues, manifest only after root growth is inhibited by exposure to toxic levels of Al in the rhizosphere [13].

Among plant species, tolerance to Al exposure varies widely, with micromolar concentrations of Al capable of producing tremendous damage to annual crops, while much higher concentrations of Al are required to generate obvious impacts on trees [14,15,16]. Even among tree species, Al can inhibit root growth (sugar maple, loblolly pine), those these impacts might be rapidly reversed when Al concentrations are reduced [17,18,19,20]. Additionally, while common crop species are known to react to Al toxicity through both Al exclusion and Al tolerance mechanisms [6,21], other means for responding to Al stress are also possible in many plant species, particularly trees [22,23,24,25,26]. For example, phenolic compounds such as terpenoids, tannins, flavonoids and alkaloids, which form strong complexes with Al ions, have been reported to participate in internal Al detoxification processes within certain trees and other Al accumulating species [27]. Given these results, elucidation of the roles filled by phenolic compounds in Al resistance processes promises to be a rewarding subject for researchers seeking to incorporate these mechanisms into breeding efforts [6,28].

As yet, Al stress is known to limit the growth and vigor of many tree species to the extent that it might be a factor determining the distribution of numerous tree species [29]. Although the physiological aspects of Al tolerance have been largely outlined over recent decades [6,30,31,32,33], the molecular machinery imparting Al tolerance remains mostly unexplored in most tree species, including for Chinese fir, which is confronted with the Al toxicity across wide areas of China, especially in South China. The relatively stable growth of Chinese fir in these acid soils suggests that this tree species has evolved mechanisms for adapting to Al toxicity. Future efforts to improve the production of Chinese fir stand to benefit by defining the range of Al tolerance possible in this species, while also cataloging the molecular processes underlying this tolerance. Monitoring gene expression responses to Al stress is one possible way to identify the genetic and molecular components responsible for Al toxicity and tolerance in Chinese fir and other tree species.

Over the last two decades, although tremendous progress has been made in understanding the physiological mechanisms of Al toxicity and tolerance in Chinese fir, the corresponding molecular mechanisms remain poorly understood. Next-generation sequencing technologies may readily access Al toxicity responses at the molecular level, which is especially valuable for species with large genomes but lacking genome sequences, such as Chinese fir (genome size is ~12 Gb). Moreover, transcriptome sequencing represents an attractive alternative to whole genome sequencing because it only analyzes transcribed portions of the genome, while avoiding non-coding and repeat sequences that can make up much of the whole genome. When observed at specific developmental stages or under controlled physiological conditions, the transcriptome can provide a complete and quantitative picture of cell transcription at a given moment.

In this study, to further understand the mechanisms of Al stress tolerance in Chinese fir at the molecular level, RNA transcriptome sequencing (RNA-seq) was conducted to identify Al-associated genes in the roots of Chinese fir using the Illumina Hiseq4000 platform. Based on bioinformatics analysis of this transcriptome, pathways involved in phenolic, phenylpropanoid and flavonoid metabolism were identified as important players in Chinese fir roots subjected to Al toxicity. Furthermore, numerous transcription factors (TFs), including bHLH, C2H2, ERF, bZIP, GRAS and MYB, were identified as potential regulators of the metabolic responses through changes in transcription observed in Chinese fir roots growing under Al stress. This transcriptome sequencing of Chinese fir roots in response to Al stress complements past physiology work and may help to elucidate previously unidentified Al resistance and tolerance mechanisms, all of which might be applied in future marker-based breeding efforts geared to genetically improving tree growth in the presence of high Al^3+^ concentrations.

## 2. Materials and Methods

### 2.1. Plant Materials and Growth Conditions

One-year-old seedlings of Chinese fir were provided by the State Forestry Administration Engineering Research Center of Chinese Fir. Firstly, the seedlings were transplanted into 0.5 mmol/L CaCl_2_ solution (pH = 4.0) and grown 7 days; and then the uniform seedlings were transplanted into 0.5 mmol/L CaCl_2_ with or without 1 mmol/L AlCl_3_ (pH = 4.0) solution, respectively. The hydroponics experiment was carried out in a temperature controlled room at 25 ± 3 °C, with the photoperiod set to 12 h light and 12 h dark cycles, constant relative humidity of 60%, 2000 lux of light intensity during the day, and continuously aerated growth media. Each treatment was observed in three biological replicates. Four cm root tips were sampled at 1 h and 72 h after initiation of Al treatments, which consisted of exposing roots to growth media (pH = 4.0) with (Al) or without (control, CK) 1 mmol/L AlCl_3_ added. Root samples weighing 0.1 g were individually inserted into 2.0 mL RNase/DNase-free tubes containing two sterile 5 mm steel balls, and immediately frozen in liquid nitrogen. All samples were stored at −80 °C prior to extracting RNA.

### 2.2. RNA Extraction, mRNA-seq Library Construction and Sequencing

Total RNA was isolated from root samples using the RNApre Pure Plant Kit (TIANGEN, Beijing, China) according to the manufacturer’s instructions. The quality and concentration of extracted RNA were determined on the Agilent 2100 Bio-analyzer (Agilent, Santa Clara, CA, USA) and NanoDrop 2000 spectrophotometer (NanoDrop Technologies, Wilmington, DE, USA), respectively.

After total RNA was extracted, mRNA was enriched using Oligo (dT) beads, while rRNA was removed using the Ribo-Zero^TM^ Magnetic Kit (Epicentre, Madison, WI, USA). Enriched mRNA was then fragmented into short fragments using fragmentation buffer prior to being reverse transcribed into cDNA using random primers. Second-strand cDNA was synthesized by DNA polymerase I, RNase H, dNTP and buffer. Then, cDNA fragments were purified with 1.8x Agencourt AMPure XP Beads, end repaired, lengthened with poly(A) tails, and ligated to Illumina sequencing adapters. Ligation products were size selected in agarose gel electrophoresis, PCR amplified, and sequenced using the Illumina HiSeqTM 4000 platform as operated by Gene Denovo Biotechnology Co. (Guangzhou, China).

### 2.3. De Novo Assembly and Functional Annotation

After removing adapters and low-quality reads, and filtering contaminant rRNA reads, clean reads were de novo assembled in the short read assembling program Trinity [34]. Expression of unigenes was calculated and normalized to RPKM (Reads Per kb per Million reads) values [35]. To annotate these unigenes, sequences were subjected to BLASTx (http://www.ncbi.nlm.nih.gov/BLAST/) searching of four databases, including the NCBI non-redundant protein (Nr) database (http://www.ncbi.nlm.nih.gov), the Swiss-Prot protein database (http://www.expasy.ch/sprot), the Kyoto Encyclopedia of Genes and Genomes (KEGG) database (http://www.genome.jp/kegg), and the COG/KOG database (http://www.ncbi.nlm.nih.gov/COG). All BLASTx searches were conducted with an E-value threshold of 10^−5^.

### 2.4. Differentially Expressed Gene (DEG) Analysis

To identify differentially expressed genes across samples or groups, the edgeR package (http://www.r-project.org/) was used, and raw counts were inputted in edgeR. Transcripts responding to Al stress with a fold change ≥ 2 and a false discovery rate (FDR) ≤ 0.05 were considered differentially expressed. All of the DEGs identified in groups TC1 (control roots at 72 h relative to control roots at 1 h) and TC2 (Al treated roots at 72 relative to Al treated roots at 1 h) were further annotated and analyzed using GO functional annotation, KEGG pathway analysis and pathway visualization in MapMan (https://mapman.gabipd.org/mapman-download).

### 2.5. Transcription Factor Analysis

The predicted plant protein coding sequences of assembled unigenes were aligned by BLASTp to Plant TFdb (http://planttfdb.cbi.pku.edu.cn/) sequences with potential TFs filtered to those matching database sequences with an E-value threshold of 10^−5^.

### 2.6. Quantitative Real-Time PCR (qRT-PCR)

Twelve RNA samples were reverse transcribed into cDNA using the reverse transcript reagent TransScript^®^ All-in-One First-Strand cDNA Synthesis SuperMix for qPCR (One-Step gDNA Removal) (TransGen Biotech, Beijing, China) following the manufacturer’s protocol. The primers used in qRT-PCR for validation of DEGs are listed in (Appendix A, SRA accession code: PRJNA577561). qRT-PCR was carried out on the LightCycler^®^ Nano Real-Time PCR System (Roche Diagnostics GmbH, Mannheim, Germany) under the following conditions: 94 °C for 30 s, and 40 cycles of 95 °C for 5 s, 58 °C for 15 s, and 72 °C for 10 s, followed by melting curve generation (60 °C to 95 °C), and a final step of 94 °C for 30 s. All treatments included three biological replicates, and the analysis of gene expression was calculated using 2^-△△Cq^ values.

## 3. Results

### 3.1. RNA Sequencing, Assembly and Functional Annotation 

To understand potential Al tolerance mechanisms active in Chinese fir roots, 12 root samples from control or Al stress treatments and collected 1 h or 72 h after transferring to treatment media were sequenced on the Illumina HiSeqTM 4000 platform. After removing adapters, low-quality reads and contaminant rRNA reads, the remaining high-quality RNA-seq datasets contained 5.3–10.4 Gb of clean data from each root sample (Appendix A). The Q20, Q30 and GC content values were 98.55–98.83%, 95.53–96.45% and 44.00–45.95%, respectively (Appendix A).

Since no reference genome is available for Chinese fir, transcriptome de novo assembly was carried out in the short read assembly program Trinity [34]. This yielded a total of 105,732 unigenes with an average length of 839 bp and an N50 length of 1411 bp.

Running BLASTx searches with an E-value threshold of 1E^-5^ led to annotation of 58,362 unigenes, including 56,577 with matches in the NCBI non-redundant (Nr) protein database, 46,225 matching Swiss-Prot protein database entries, 39,868 with matches in the eukaryotic Orthologous Groups (KOG) database, and 26,914 unigenes aligning with sequences in the Kyoto Encyclopedia of Genes and Genomes (KEGG) pathway database (Appendix A). About 22.35% (23,632/105,732) of the assembled unigenes could be assigned to homologs in all four databases (Figure 1A). Based on the Nr database, 28.41% of the assembled unigenes exhibited homology (10^−20^ < E-value <= 10^−5^) to plant proteins, while 47.40% showed strong homology (10^−100^ < E-value <= 10^−20^) and the remaining 24.19% displayed very strong homology (E-value <10^−100^) to available plant sequences (Appendix A). Determination of species represented in matches with assembled unigenes (Appendix A) in the Nr database revealed that a high percentage of Chinese fir transcripts closely matched sequences of Anthurium amnicola (16.79%), Amborella trichopoda (6.33%), Nelumbo nucifera (4.37%), Nannochloropsis gaditana (2.76%), Picea sitchensis (2.21%), Klebsormidium flaccidum (2.04%), Ectocarpus siliculosus (1.94%), Elaeis guineensis (1.69%), Vitis vinifera (1.58%), and Phoenix dactylifera (1.42%). 

GO annotations were assigned to unigenes in the Blast2GO [36] application using the Nr annotation results from the BLASTx alignments described above, and functional classification of unigenes was performed using WEGO [37] software. A total of 10,837 Unigenes were therefore categorized into 44 main groups distributed among GO biological process, molecular function, and cellular component categories (Figure 1). As shown in Figure 1, the top five most represented metabolic process subcategories in the Chinese fir transcriptome were metabolic process, cellular process, single-organism process, response to stimulus, and localization. The top five most represented cellular component subcategories were cell, cell part, organelle, macromolecular complex, and membrane, and the top five most represented molecular function subcategories were catalytic activity, binding, transporter activity, structural molecule activity, and nucleic acid binding transcription factor activity.

To place the transcripts identified in this study in a metabolic context, BLASTx hits from the Nr database obtained for the Chinese fir root tip transcriptome were also used to assign unigenes to KEGG pathways. In this analysis, 15,542 of the 26,914 identified unigenes mapped to 137 KEGG pathways (Appendix A). The top four KEGG pathway categories based on the number of assigned unigenes were Metabolic pathways (ko01100, 38.12% of unigenes), Biosynthesis of secondary metabolites (ko01110, 21.12%), Ribosome (ko03010, 11.85) and Biosynthesis of antibiotics (ko01130, 10.58%).

The 56,577 unigenes with hits in the Nr database were also queried for homology with sequences in the KOG database to further assess the validity and integrity of the transcriptome sequences, as well as, and to further classify potential functions in Chinese fir. In this analysis, a total of 39,868 unigenes were assigned to 25 KOG classifications (Figure 2, Appendix A). Among them, the top five KOG classifications were R (General function prediction only, 13,154 unigenes), O (Posttranslational modification, protein turnover, chaperones, 8795 unigenes), T (Signal transduction mechanisms, 8591 unigenes), J (Translation, ribosomal structure and biogenesis, 4493 unigenes) and A (RNA processing and modification, 4202 unigenes).

### 3.2. Analysis of Differential Expressed Genes (DEGs) 

The edgeR package (http://www.r-project.org/) was used to identify differentially expressed genes (DEGs) between control and Al treated roots. Transcripts with at least a 2-fold change in RPKM expression values and a false discovery rate (FDR ≤ 0.05) corrected t-test *p*-value corr in pairwise comparisons were considered significant DEGs. These DEGs were then subjected to enrichment analysis of GO categories and KEGG pathways. Only 3 up-regulated DEGs were identified in the T1 group (Al-1 h/ CK-1 h), while 140 up-regulated DEGs and 6244 down-regulated DEGs were identified in the T2 group (Al-72 h/ CK-72 h) (Figure 3). In time course comparisons, there were 3918 up-regulated DEGs and 396 down-regulated DEGs in the TC1 group (CK-72 h/ CK-1 h), while 3130 up-regulated DEGs and 20138 down-regulated DEGs were identified in the TC2 group (Al-72 h/Al-1 h) (Appendix A). The number of DEGs was notably high in the TC2 group, especially for down-regulated DEGs. To better understand the potential mechanisms of Chinese fir root responses to Al, the TC1 and TC2 grouped DEGs were subjected to further analysis. 

Interestingly, 785 DEGs were identified in both TC1 and TC2 comparisons (Figure 3B, Appendix A), the majority displayed consistent expression patterns between comparisons. However, eight DEGs did display altered expression patterns between TC1 and TC2. These eight unigenes with altered expression patterns between TC1 and TC2 were Unigene0048758 (no annotation), Unigene0073541 (fatty acid synthase), Unigene0071412 (Sulfite reductase), Unigene0057949 (Inositol-3-phosphate synthase), Unigene0002857 (acetyl-coenzyme a synthetase), Unigene0063838 (chaperone, heat shock protein 70), Unigene0042239 (glutathione S-transferase theta-1), and Unigene0060101 (uroporphyrin-III C-m). The majority of DEGs with consistent expression patterns observed between TC1 and TC2 might not be affected by Al, but rather could have expression altered as part of normal growth and development processes in Chinese fir roots.

### 3.3. GO Analysis of DEGs

To determine the main biological functions filled by TC1 and TC2 DEGs, these transcripts were mapped to GO terms in BLAST2GO analysis.

In the TC1 group, a total of 4314 DEGs, including 3918 up-regulated and 396 down-regulated DEGs, were categorized to 35 GO terms (Figure 4, Appendix A). Among them, the top 3 biological process GO terms were metabolic process (536 up-regulated DEGs, 18 down-regulated DEGs), cellular process (352 up-regulated DEGs, 13 down-regulated DEGs), and single-organism process (295 up-regulated DEGs, 17 down-regulated DEGs). The top 4 cellular component GO terms were cell (480 up-regulated DEGs, 8 down-regulated DEGs), cell part (480 up-regulated DEGs, 8 down-regulated DEGs), organelle (282 up-regulated DEGs, 7 down-regulated DEGs), and macromolecular complex (256 up-regulated DEGs). The top 2 molecular function GO terms were catalytic activity (416 up-regulated DEGs, 24 down-regulated DEGs), and binding (289 up-regulated DEGs, 16 down-regulated DEGs).

In the TC2 group, a total of 23,268 DEGs, including 3130 up-regulated and 20,138 down-regulated DEGs, were categorized to 41 GO terms. Among them, the top 3 biological process GO terms were metabolic process (287 up-regulated DEGs, 1296 down-regulated DEGs), cellular process (197 up-regulated DEGs, 1097 down-regulated DEGs), and single-organism process (163 up-regulated DEGs, 912 down-regulated DEGs). The top 4 cellular component GO terms were cell (307 up-regulated DEGs, 948 down-regulated DEGs), cell part (307 up-regulated DEGs, 948 down-regulated DEGs), organelle (221 up-regulated DEGs, 613 down-regulated DEGs), and macromolecular complex (175 up-regulated DEGs, 322 down-regulated DEGs). The top 2 molecular function GO terms were catalytic activity (198 up-regulated DEGs, 1351 down-regulated DEGs), and binding (172 up-regulated DEGs, 833 down-regulated DEGs) (Appendix A).

### 3.4. KEGG Pathway Analysis of DEGs 

To better understand the metabolic functions of DEGs identified in Chinese fir roots under Al stress, TC1 and TC2 group DEGs were queried against the KEGG database.

TC1 DEGs mapped into 127 KEGG pathways (Figure 5). The top five KEGG pathways based on the number of assigned DEGs were Global and overview maps (54 down-regulated DEGs, 526 up-regulated DEGs), Translation (1 down-regulated DEGs, 422 up-regulated DEGs), Carbohydrate metabolism (13 down-regulated DEGs, 182 up-regulated DEGs), folding, sorting and degradation (4 down-regulated DEGs, 187 up-regulated DEGs), and amino acid metabolism (13 down-regulated DEGs, 164 up-regulated DEGs).

TC2 DEGs, most of which were down-regulated, mapped into 134 KEGG pathways (Figure 5, Appendix A). The KEGG pathway Environmental adaptation is particularly notable for including 83 down-regulated and 18 up-regulated TC2 DEGs, as well as 3 down-regulated and 15 up-regulated TC1 DEGs. These changes in Environmental adaptation pathways may include important actors filling roles in Chinese fir root responses to Al stress.

Enrichment analysis of KEGG terms associated with DEGs was conducted to further understand potential metabolic roles played by these gene products in Chinese fir roots subjected to Al stress treatments for 1 h and 72 h. TC1 DEGs were significantly enriched in Citrate cycle, Lysine biosynthesis, Oxidative phosphorylation, and specific amino acid biosynthesis and metabolism pathway, while TC2 DEGs were significantly enriched in different pathways, including Flavone and flavonol biosynthesis, Terpenoid backbone biosynthesis, Steroid biosynthesis, Phenylpropanoid biosynthesis, and Linoleic acid metabolism (Figure 6, Appendix A). These differences in KEGG pathway enrichment between TC1 and TC2 might mark pathways that are important in Chinese fir root responses to Al stress. 

The software MapMan was used to display the global overview of DEGs in Chinese fir roots responding to Al stress. This visualization helps to understand the contrasting roles between primarily up-regulated TC1 DEGs (Figure 7A) and predominantly down-regulated in TC2 DEGs (Figure 7B). Notably, contrasting DEG responses between TC1 and TC2 comparisons were evident in primary metabolism pathways, including those associated with trehalose, callose, phospholipid synthesis and steroids (sphingolipids), FA synthesis and desaturation, glycolysis, the tricarboxylic acid (TCA) cycle, starch, sucrose, ascorbate and glutathione metabolism (Figure 7). Among the analyzed DEGs, six encoding enzymes putatively involved in trehalose biosynthesis were up-regulated over time in control roots group (Figure 7A), while 10 such DEGs were down-regulated over time in Al treated roots (Figure 8B). Six other DEGs encoding GSL10, a member of the Glucan Synthase-Like (GSL) family believed to be involved in the synthesis of the cell wall component callose were up-regulated from 1 h to 72 h in control roots (TC1), while 4 glycosyl transferase DEGs, possibly acting in callose synthase or glucan synthase-like 7 (GSL7) reactions, were down-regulated from 1 h to 72 h in Al stress-treated roots (TC2). Furthermore, 18 DEGs involved in cellulose biosynthesis, 7 DEGs of FASCICLIN-like arabinogalactan proteins putatively localized in plasma membranes, and 1 Alpha-1,4-glucan-protein synthase DEG possibly involved in plant cell wall synthesis were all down-regulated in TC2 comparisons, none of which were significant DEGs in TC1 comparisons. Moreover, two of the three DEGs encoding a putative sorbitol dehydrogenase that can be thiolated in vitro, and three of the seven DEGs annotated as raffinose family proteins possibly acting as galacturonosyltransferases were down-regulated in TC2, but were unaffected in TC1 comparisons. Finally, except for down-regulation of three chloroplast localized DEGs annotated as phosphofructokinase 3 (PFK3), the remaining 27 DEGs associated with the energy payoff phase of the glycolytic pathway were all upregulated in TC1 roots, including DEGs encoding triosephosphate isomerase, phosphoenolpyruvate carboxylase-related kinase 1 (PEPKR1), enolase, and pyruvate kinase were all up-regulated in TC1 comparisons. In contrast, 63 of the 66 similarly annotated DEGs in TC2 were down-regulated.

This contrast of DEG response patterns was also observed between TC1 and TC2 comparisons in analysis of DEGS associated with secondary metabolism, including DEGs annotated to Phenylpropanoid and Phenolic biosynthesis, Flavonoid biosynthesis, Amino acid synthesis and degradation, Ammonia metabolism and Sulfur metabolism, and Nucleotide metabolism (Figure 7). Among TC1 DEGs, only eight of the 47 that were annotated as secondary metabolism pathways were down-regulated, while the remaining 39 DEGs were up-regulated. In contrast, only 53 of 342 TC2 DEGs were up-regulated, while the remaining 289 DEGs were down-regulated. Among all DEGs associated with secondary metabolism, 19 exhibited similar patterns between the TC1 and TC2 comparison groups (8 were consistently down-regulated over time and 11 were up-regulated). This indicates that these DEGs may be constitutively expressed genes in Chinese fir roots regardless of Al stress. 

Based on the KEGG pathway enrichment results for TC1 and TC2 DEGs, TC2 DEGs were found to be significantly enriched in different pathways than TC1 DEGs, with TC2 DEGs being enriched in flavone and flavonol biosynthesis, terpenoid backbone biosynthesis, steroid biosynthesis, phenylpropanoid biosynthesis, and linoleic acid metabolism (Figure 6, Appendix A). Therefore, further analysis was conducted only for TC1 and TC2 DEGs involved in phenylpropanoid and phenolic biosynthesis and flavonoid biosynthesis, which are highlighted in light blue in Figure 7, and are presented in more detail in Figure 8 and Figure 9.

Phenylpropanoid biosynthesis and flavonoid biosynthesis are both known to play important roles in plant abiotic stress and biotic stress responses. In this study, several DEGs were annotated as involved in phenylpropanoid biosynthesis and metabolism, including phenylalanine ammonia-lyase (*PAL*), 4-coumarate-CoA ligase (*4CL*), cinnamic acid 4-hydroxylase (*C4H*), cinnamoyl-CoA reductase (*CCR*), cinnamyl alcohol dehydrogenase (*CAD*), 4-coumarate 3-hydroxylase (*C3H*), and caffeic acid 3-O-methyltransferase (*COMT*). Similarly, several DEGs possibly involved in flavonoid biosynthesis were also observed, including DEGs annotated as chalcone synthase (Naringenin-chalcone synthase) (*CHS*), flavonol synthase (*FLS*), dihydroflavonol-4-reductase (*DFR*), anthocyanidin reductase (*ANR*), leucoanthocyanidin reductase (*LAR*), and caffeoyl-CoA O-methyltransferase (*CCOAOMT*). Among them, *4CL*, *CHS*, *FLS*, *ANS*, and *CAD* were all up-regulated in the TC1 comparison, while *CCR* was down-regulated (Figure 9). Meanwhile, expression responses for these phenylpropanoid and flavonoid associated genes were in the opposite direction in TC2 DEGs compared to TC1 DEGs, with the exception that *CCR* responded similarly in both comparison groups. As a result, it is reasonable to conclude that DEGs involved in phenylpropanoid and flavonoid metabolism may play important roles in Chinese fir roots in response to Al stress.

### 3.5. Transcription Factor Analysis

MapMan was also used to visualize TC1 and TC2 DEGs associated with regulatory terms, such as transcription factor, protein modification, protein degradation, hormone metabolism, signaling pathway, and redox pathway. This visualization of differentially expressed regulatory proteins is shown for TC1 and TC2 comparisons in Figure 10. In this study, we focused on TF analysis of TC1 and TC2 DEGs putatively involved in regulation pathways.

A total of 1615 unigenes hit matches from 58 families in the plant transcription factor database PlantTFdb (Appendix A). The top 11 TF families associated with TC1 and TC2 DEGs were C2H2 (10.77%), ERF (10.28%), bHLH (8.79%), MYB-related (7.12%), MYB (6.75%), bZIP (5.20%), C3H (5.14%), GRAS (3.84%), Trihelix (3.34%), WRKY (2.97%) and NAC (2.85%) (Figure 11). These TF types are known to regulate the expression of many genes, including those involved in secondary metabolism, regulation of gene expression and response to biotic and abiotic stress.

The 1615 TFs identified above were then clustered into 20 profiles based on expression across treatments (Figure 12). Of these profiles, profile 9 with 305 TFs, profile 4 with 265 TFs, profile 14 with 146 TFs and profile 15 with 119 TFs were each significantly enriched compared to an expected equal distribution of observed profiles. 

To further determine which TFs may play roles in Chinese fir root responses to Al stress, the TFs of TC1 and TC2 DEGs were predicted using the plant transcription factor database (PlantTFdb) (Appendix A). In this analysis, 64 TFs from 21 TF families were identified in TC1 DEGs, and 495 TFs from 40 TF families were identified in TC2 DEGs (Figure 13). Interestingly, expression of bHLH (basic helix-loop-helix), C2H2 (Cys2-His2), ERF (ethylene response factor), bZIP, GRAS, and MYB (Myeloblastosis) TF family members all changed considerably in TC2 DEGs, indicating that these TFs might be involved in Chinese fir root responses to Al stress (Figure 13, Appendix A). 

### 3.6. Validation of RNA-Seq Results by qRT-PCR

To validate the accuracy of RNA-Seq, qRT-PCR was conducted to measure the expression of the 21 selected DEGs, including one Al-activated malate transporter 10-like protein (Unigene0062724), two auxin/Al responsive-like proteins (Unigene0067300 and Unigene0067301), two peroxidase-like proteins (Unigene0007466 and Unigene0075814), five phosphate transporters (Unigene0067425, Unigene0007397, Unigene0074352, Unigene0074941, Unigene0012553), and 10 DEGs involved in flavonoid and phenylpropanoid pathways (Unigene0094771, Unigene0070938, Unigene0076509, Unigene0071623, Unigene0070437, Unigene0063962, Unigene0072773, Unigene0022972, Unigene0069446, Unigene0068571) (Figure 14). The results indicate that the expression patterns of these selected DEGs were all similar between RNA-Seq and qRT-PCR observations.

## 4. Discussion

In this study, transcriptome analysis was employed to investigate mechanisms of Al tolerance in Chinese fir. As such, this study constitutes the first investigation of transcriptomic analysis of Chinese fir root responses to Al toxicity. After subjecting Chinese fir roots to 1 mM Al and control treatments for 1 h or 72 h, RNA-seq analysis returned a total of 105,732 unigenes, with an average length of 839 bp and an N50 length of 1411 bp. Compared with previous studies of transcriptomes obtained from Chinese fir vascular cambium [38] or immature xylem [2] tissue sampled at different developmental phases, the sequencing and assembly conducted herein returned much longer N50 (1411 bp) and mean (839 bp) lengths. This demonstrated that the final assembly quality in our study was satisfactory, and further suggests that transcriptome data can provide accurate sequence data for future cloning and functional analysis efforts aimed to improve Al resistance and tolerance in cultivated Chinese fir. 

Queries of four sequence databases with the 105732 assembled unigenes returned annotations for 58,362 of them, of which 56,577 were homologous with sequences in the Nr database. To maximize the possible functional predictions of transcripts isolated from Chinese fir roots responding to Al stress, further analysis was limited to these 56,577 unigenes aligning to Nr annotated sequences. Furthermore, DEG analysis was also limited to this set of 56,577 unigenes. Upon exposure to 1 mM Al for 1 h or 72 h, Chinese fir roots yielded only 3 DEGs in the T1 comparison group, but 6384, 4314 and 23,268 DEGs in T2, TC1 and TC2 comparison groups, respectively (Figure 3A). The fact that only 3 DEGs were found in the T1 comparison, to some extent, further illustrates that Chinese fir is a relatively resistant to Al, and short time Al treatment could not induce large changes in gene expression. However, with the extension of Al treatment time, the number of down-regulated DEGs increased significantly, as evident in TC2 comparison. This suggests that rapid responses to Al toxicity in Chinese fir roots might be dominated by down-regulation of many genes. Further analysis of TC1 and TC2 DEGs also revealed that TC2 DEGs were significantly enriched in different pathways than TC1 DEGs. Significant pathways of interest associated with TC2 DEGs include flavone and flavonol biosynthesis, as well as phenylpropanoid biosynthesis (Figure 7), which suggests that these two pathways might participate in Chinese fir root responses to Al stress. Coincidently, Chakraborty et al. reported that phenolic and flavonoid contents in root exudates of Azolla increased significantly under exposure to 100, 250, and 500 µM Al stress treatments, but declined in the highest exposure treatment of 750 µM Al [39]. These research findings appear to support our conclusion that DEGs involved in phenylpropanoid and flavonoid pathways are largely down-regulated in Chinese fir roots under exposure to 1 mM Al stress. Moreover, Kidd et al. also reported that phenolics and flavonoids can chelate with Al in an Al-tolerant maize cultivar [40]. Thus, our results confirm existing evidence that phenolics and flavonoids play important roles in Chinese fir root responses to Al toxicity. However, the specific secondary metabolites involved in responses to Al stress, and the mechanisms through which they act need to be studied in more detail in future research.

In this study, we also identified hundreds of Al responsive transcription factors belonging to diverse families, including bHLH, C2H2, ERF, bZIP, and MYB members identified among TC2 DEGs, along with other, largely distinct sets of TFs identified among TC1 DEGs. These results indicate that the TFs found among TC2 DEGs might be involved in Chinese fir root responses to Al toxicity (Figure 14). 

All of the TF families illuminated in this study are known to participate in plant abiotic and biotic stress responses [41]. In a recent study, the C2H2-type TF STOP1 was demonstrated as a crucial player in Arabidopsis resistance responses to Al stress through its role in inducing the expression of a set of Al-resistance related genes, including *AtALMT1* [42]. In addition, several previous studies have also reported that MYB and bHLH TFs can modulate flavonoid biosynthesis in plants experiencing Al stress. Yet, in this study, none of the identified TFs are known to be involved in phenylpropanoid or flavonoid pathways. Whether any of these identified TFs interacts with the expression of genes participating in either of these two pathways should also be worthwhile subjects of future research projects.

Several groups have reported that ERF Tfs regulate many key functions, including developmental processes and responses to abiotic and biotic stresses [43,44]. In this study, the TF family containing the largest number of TC2 DEGs was the ERF family (79), which may induce the expression of genes acting in the synthesis of phenols in secondary metabolic pathways. These findings suggested that ERF TFs might play important roles in Chinese fir root responses to Al toxicity. Deciphering the mechanisms through which these TFs act might prove to be a key component of future research meant to elucidate and improve Chinese fir root responses to Al toxicity.

## 5. Conclusions

Little is known about the molecular mechanisms underlying Chinese fir responses to Al stress. This study is the first attempt to investigate changes of Chinese fir roots in response to Al toxicity at the transcriptional level, which also provided indications of which biochemical mechanisms underlie Al tolerance in Chinese fir. Several DEGs associated with phenylpropanoid and flavonoid metabolism were identified in this research as strongly down-regulated in Chinese fir roots responding to Al toxicity, suggesting that phenylpropanoid and flavonoid metabolism could participate in regulating responses to Al toxicity in Chinese fir. In addition, bHLH, C2H2, ERF, bZIP, MYB and the other TFs were differentially expressed among Al treatments over time, and, therefore, may play important roles in Chinese fir root responses to Al stress.

Overall, this research represents a first step toward understanding the molecular mechanisms behind Chinese fir response to Al stress. The transcriptome data in this study will help to further develop novel Al-resistance strategies in Chinese fir and other tree species.

## Figures and Tables

**Figure 1 genes-10-00835-f001:**
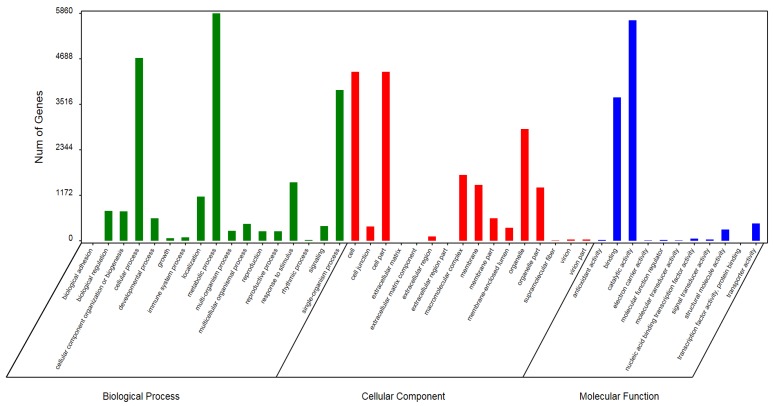
Gene Ontology (GO) classification of all unigenes identified from transcriptomes of Chinese fir root tips exposed to Al stress conditions. The results are summarized in three main categories, biological process, cellular component and molecular function.

**Figure 2 genes-10-00835-f002:**
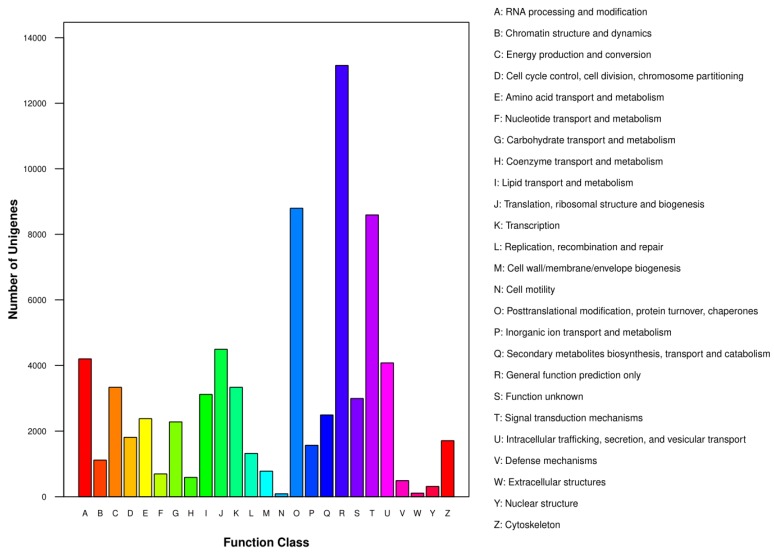
Eukaryotic ortholog group (KOG) function classification of Chinese fir root tip transcripts identified in an Al stress experiment. The X-axis lists KOG clusters, and the Y-axis is the number of unigenes assigned to each KOG classification.

**Figure 3 genes-10-00835-f003:**
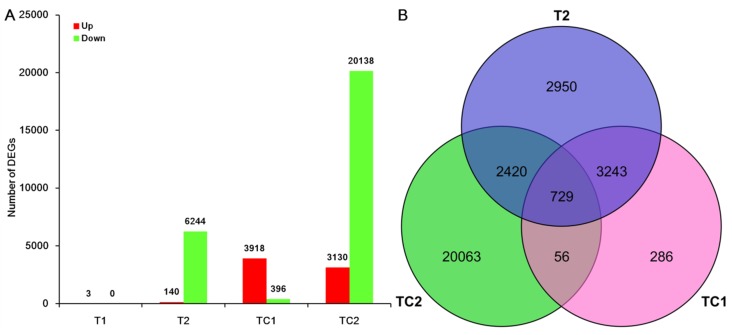
Summary of differentially expressed genes (DEGs) in Chinese fir roots under Al stressed. (**A**) The X-axis lists comparison groups, T1: indicated the DEGs number in the group of Al-1 h compared with CK-1 h, Al-1 h/CK-1 h; T2: indicated the DEGs number in the group of Al-72 h compared with CK-72 h, Al -72 h/CK-72 h; TC1: CK-72 h/ CK-1 h, TC2: Al-72 h /Al-1 h. The Y-axis is the number of up-regulated or down-regulated DEGs in each comparison group. (**B**) Venn diagram of DEGs identified in three differential comparison groups.

**Figure 4 genes-10-00835-f004:**
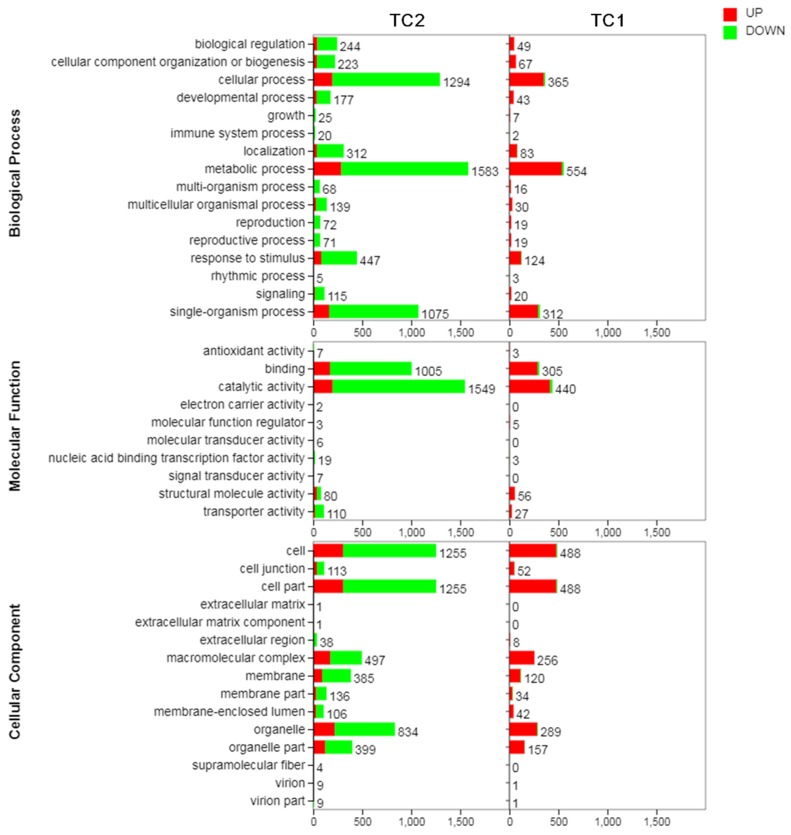
GO classifications of DEGs identified in Chinese fir roots subjected to Al stress treatments. TC1: CK-72 h/ CK-1 h. TC2: Al-72 h /Al-1 h. DEGs were annotated in 3 GO categories: biological process, molecular function, and cellular component. Red represents up-regulated DEGs. Green represents down-regulated DEGs. The number besides each bar represents the number of DEGs associated with the corresponding GO term.

**Figure 5 genes-10-00835-f005:**
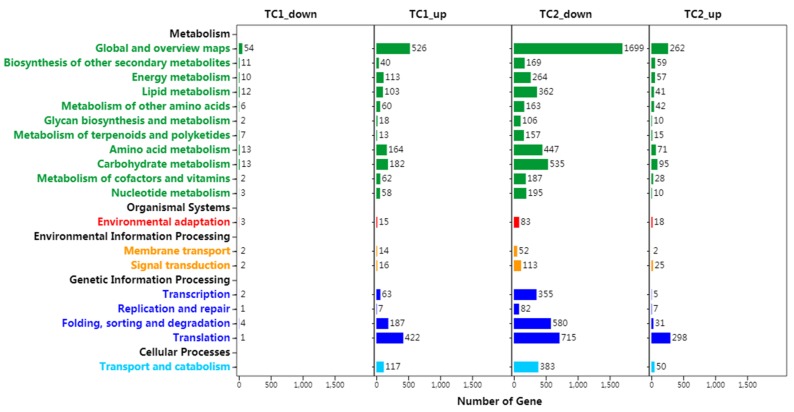
**Kyoto Encyclopedia of Genes and Genomes** (KEGG) pathway annotations of DEGs identified in Chinese fir roots subjected to Al stress. TC1_up: up-regulated DEGs in the comparison of CK-72 h with CK-1 h; TC1_down: down-regulated DEGs in the comparison of CK-72 h with CK-1 h; TC2_up: up-regulated DEGs in the comparison of Al-72 h with Al-1 h; TC2_down: down-regulated DEGs in the comparison of Al-72 h with Al-1 h. Categories in black font on the Y-axis are KEGG B classes, while categories in other colored fonts on the Y-axis are specific pathways classified under each KEGG B class. The X-axis represents the number of DEGs annotated to each pathway.

**Figure 6 genes-10-00835-f006:**
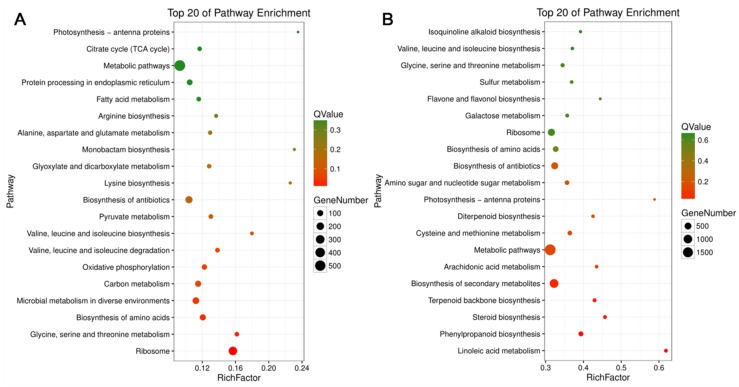
KEGG pathway analysis of TC1 and TC2 grouped DEGs. (**A**) Top 20 enriched pathways associated with TC1 DEGs (CK-72 h/ CK-1 h); (**B**) top 20 enriched pathways associated with TC2 DEGs (Al-72 h /Al-1 h). The Rich Factor refers to the ratio of the number of DEGs to the total number of genes in the pathway. The larger the rich factor, the higher the Rich Factor. The size of the bubbles indicates the number of DEGs, with larger bubbles indicating the more pathway DEGs. Bubble color indicates the level of significance (q-value) in FDR corrected t tests, with deeper red color indicating higher significance of differential expression.

**Figure 7 genes-10-00835-f007:**
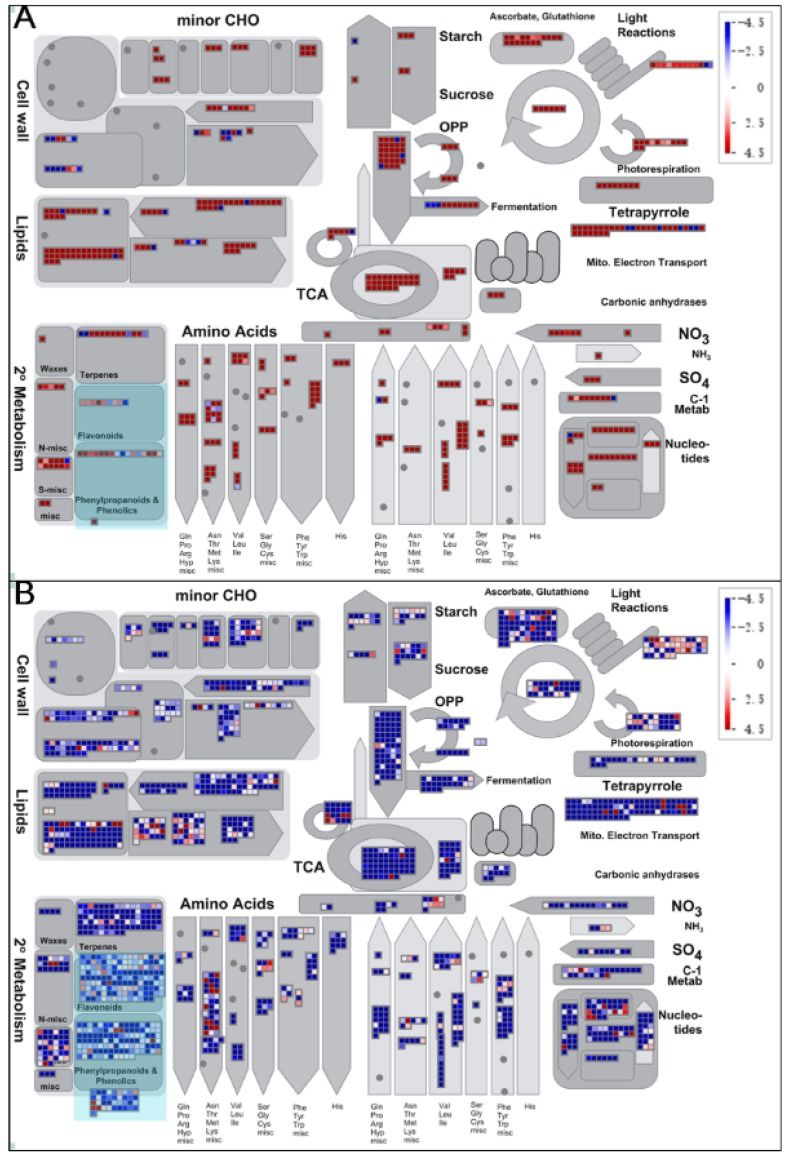
Metabolic overview of DEG identified in Chinese fir roots responding to Al stress. This MapMan generated global overview of DEGs in a metabolic pathway context showing transcriptional changes observed in Chinese fir roots subjected to control or Al treatments for 1 h and 72 h. (**A**) TC1 DEGs (log_2_(CK-72 h/CK-1 h)). (**B**) TC2 DEGs (log_2_(Al-72 h/Al-1 h). Deeper blue indicates stronger down-regulation, and deeper red indicates more up-regulation. Individual DEGs are represented by small squares. The scale from −4.5 to +4.5 represents the normalized log_2_ of fold change in each compared group.

**Figure 8 genes-10-00835-f008:**
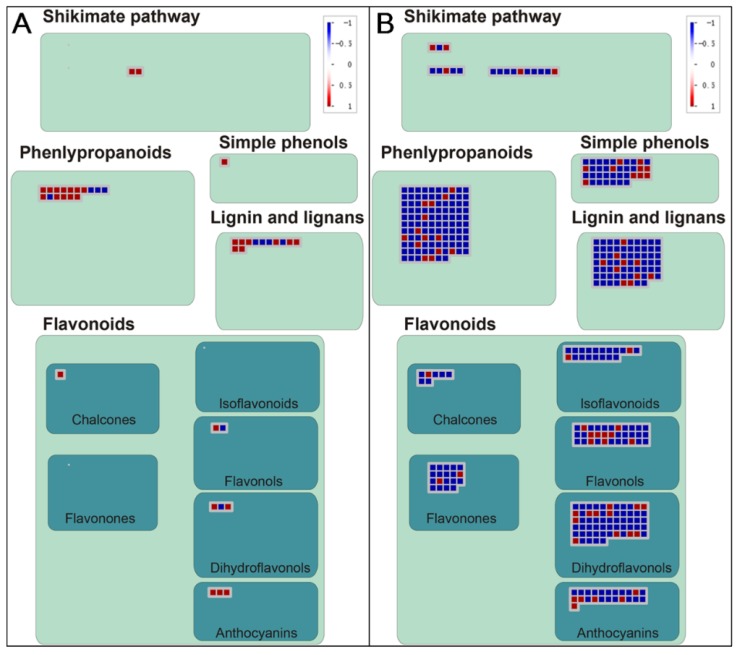
Visualization of DEGs associated with phenylpropanoid and flavonoid metabolism. The TC1 and TC2 DEGs included here are associated with the biosynthesis of shikimate, phenlypropanoids, simple phenols, lignin and lignans, and flavonoids. (**A**) TC1 DEGs (log_2_(CK-72 h/CK-1 h)). (**B**) TC2 DEGs (log_2_(Al-72 h/Al-1 h). Individual DEGs are represented by small squares. Deeper blue indicates stronger down-regulation, and deeper red indicates more up-regulation. The scale from −1 to +1 represents normalized log_2_ of fold change in each comparison group.

**Figure 9 genes-10-00835-f009:**
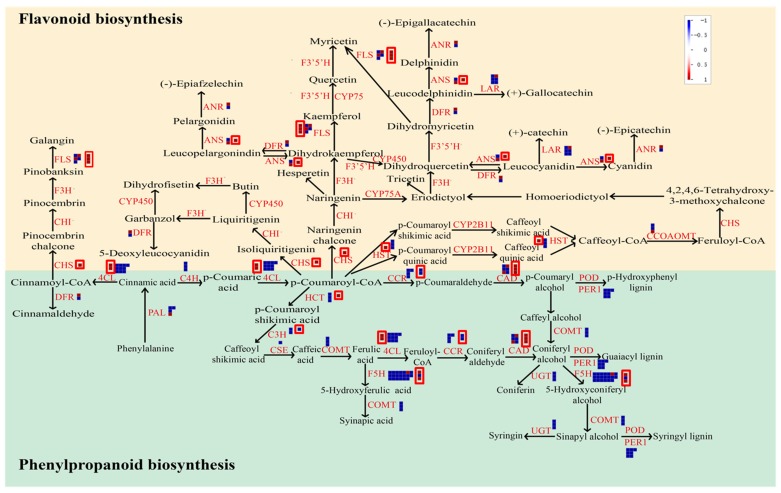
Visualization of TC1 and TC2 DEGs involved in phenylpropanoid and flavonoid biosynthesis. Red boxes outline squares representing TC1 DEGs (log_2_(CK-72 h/CK-1 h)), and squares lacking a red outline box represent TC2 DEGs (log_2_(Al-72 h/Al-1 h). Within squares, deeper blue indicates stronger down-regulation, and deeper red indicates more up-regulation. Flavonoid biosynthesis is highlighted by a light red background, and phenylpropanoid biosynthesis is highlighted by a light blue background. PAL: phenylalanine ammonia-lyase; 4CL: 4-coumarate-CoA ligase; C4H: cinnamic acid 4-hydroxylase (CYP73A: trans-cinnamate 4-monooxygenase, (EC:1.14.13.11)); CCR: cinnamoyl-CoA reductase; CAD: cinnamyl alcohol dehydrogenase, (EC:1.1.1.195); C3H: 4-coumarate 3-hydroxylase (C3′H: coumaroylquinate (coumaroylshikimate) 3′-monooxygenase, (EC:1.14.13.36)); CSE: caffeoyl shikimate esterase; COMT: caffeic acid 3-O-methyltransferase; HCT (HST): shikimate O-hydroxycinnamoyltransferase, (EC:2.3.1.133); CCOAOMT: caffeoyl-CoA O-methyltransferase, (EC:2.1.1.104); CHS: chalcone synthase (Naringenin-chalcone synthase), (EC:2.3.1.74); FLS: flavonol synthase, (EC:1.14.11.23); F3’5’H (CYP75A): flavonoid 3′,5′-hydroxylase, (EC:1.14.13.88); F3’H: flavonoid 3′-monooxygenase, (EC:1.14.13.21); DFR: dihydroflavonol-4-reductase, (EC:1.1.1.219); F3H: naringenin 3-dioxygenase, (EC:1.14.11.9); LAR: leucoanthocyanidin reductase; ANR: anthocyanidin reductase, (EC:1.3.1.77); ANS: Anthocyanidin synthase; CHI: chalcone isomerase, (EC:5.5.1.6); REF1: coniferyl-aldehyde dehydrogenase; UGT (UFGT): flavonol 7-O-glucosyltransferase, (EC:2.4.1.237) or Anthocyanidin 3-O-glucosyltransferase (EC:2.4.1.115) (Flavonol 3-O-glucosyltransferase) (UDP-glucose flavonoid 3-O-glucosyltransferase) (Anthocyanin rhamnosyl transferase).

**Figure 10 genes-10-00835-f010:**
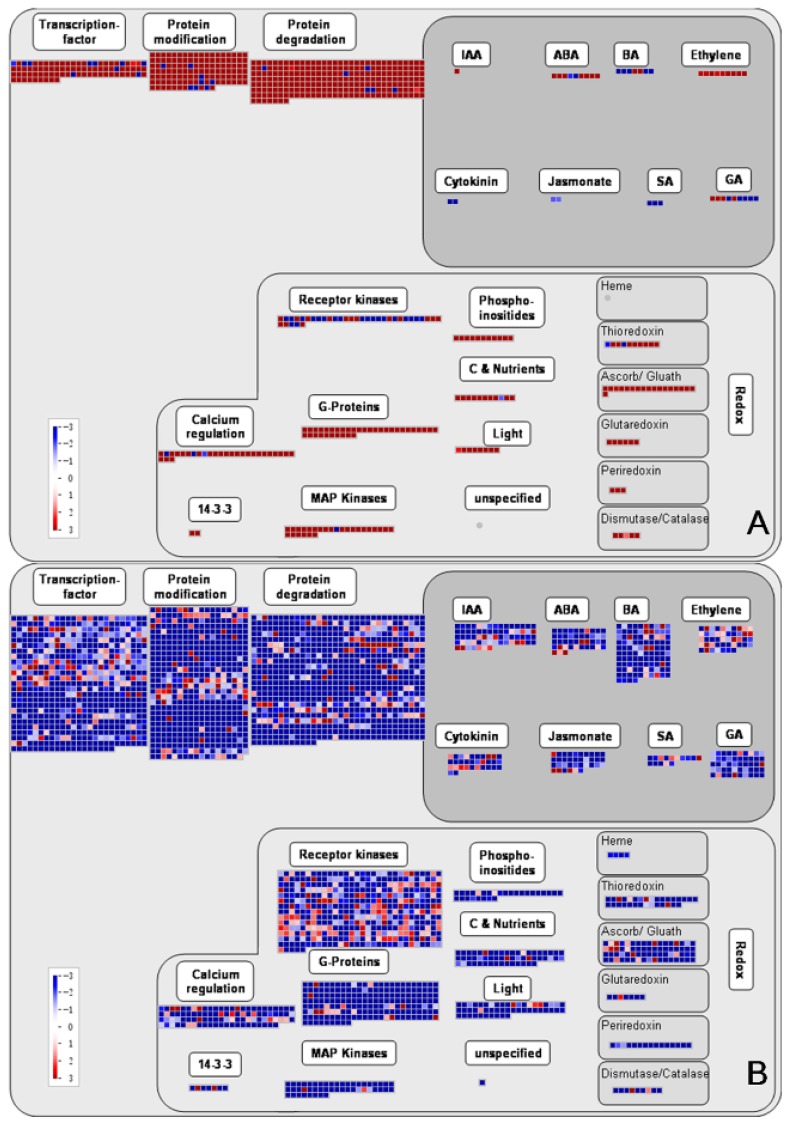
MapMan overview of regulation pathways containing TC1 or TC2 DEGs. (**A**) TC1 DEGs (log_2_(CK-72 h/CK-1 h)). (**B**) TC2 DEGs (log_2_(Al-72 h/Al-1 h). Deeper blue indicates stronger down-regulation, and deeper red indicates more up-regulation. Individual DEGs are represented by small squares. The scale from −3 to +3 represents the normalized log_2_ of fold change in each comparison group. IAA: Indole-3-acetic acid; ABA: abscisic acid; BA: brassinosteroid; SA: salicylic acid; GA: gibberellins.

**Figure 11 genes-10-00835-f011:**
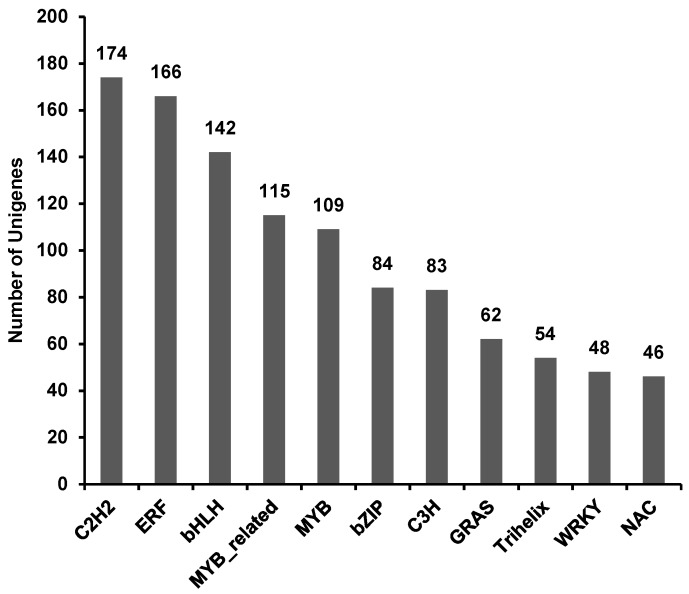
Top 11 differentially regulated TF families in Chinese fir roots responding to Al stress. The X-axis lists the top 11 TF families. The Y-axis indicates the number of Unigenes assigned to each TF family.

**Figure 12 genes-10-00835-f012:**
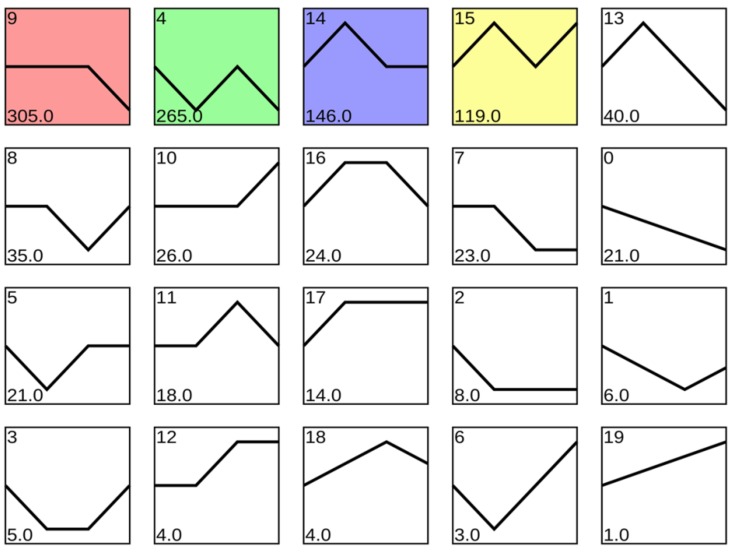
Cluster analysis of 1615 TFs differentially expressed across Al treatments. Unigenes identified as TFs were classified based on similarity of expression patterns in Al and CK treated Chinese fir roots sampled after 1 h and 72 h exposure to Al stress. Profiles are ordered based on the number of TFs. Profiles with colored backgrounds were significantly enriched compared to an expectation of equal distribution of TFs among profiles. Profiles with a white background were not significantly enriched.

**Figure 13 genes-10-00835-f013:**
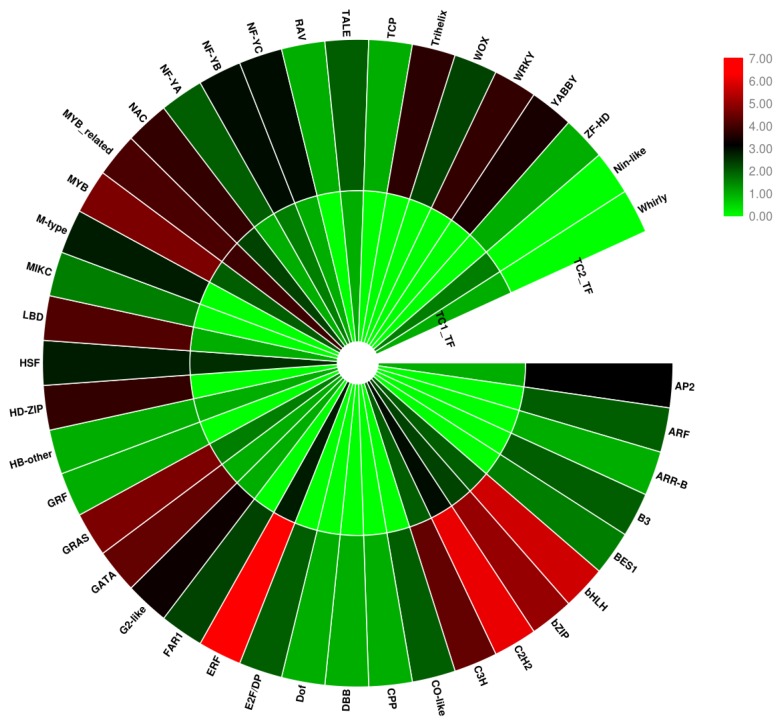
Expression patterns of DEGs identified as TFs in TC1 and TC2 DEGs. The circle legend shows the TF type, TC1_TF: TC1 DEGs identified as TFs; TC2_TF: TC2 DEGs identified as TFs. The scale 0 to 7, represents log_2_(TF number). Red color marks more abundant TFs, and green color marks less prevalent TFs.

**Figure 14 genes-10-00835-f014:**
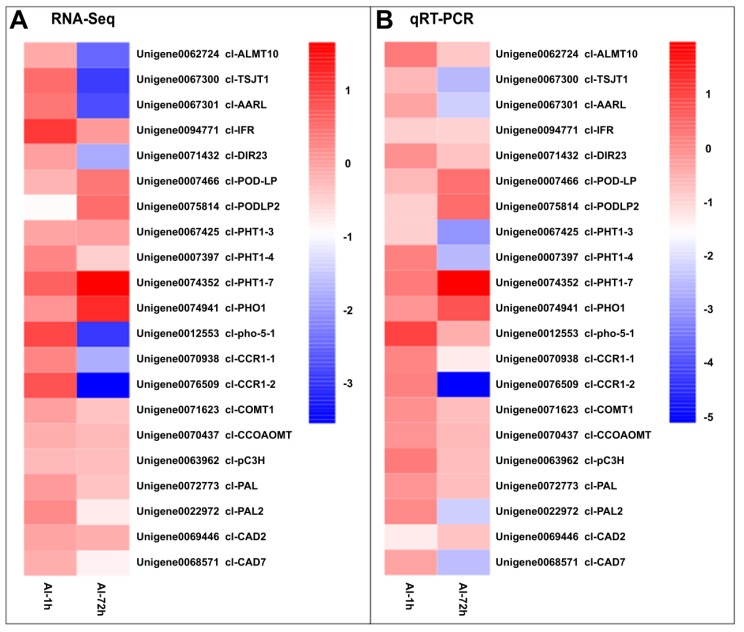
qRT-PCR validation of 21 selected DEGs. RNA-Seq and qRT-PCR data are displayed as log_2_(fold changes). A: RNA-Seq results for 21 selected DEGs. B: qRT-PCR results for 21 selected DEGs. Al1h: log_2_(Al-1 h/CK-1 h); Al72h: log_2_(Al-72 h/CK-72 h).

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
