# Peer review of "Transcriptomic Revelation of Phenolic Compounds Involved in Aluminum Toxicity Responses in Roots of Cunninghamia lanceolata (Lamb.) Hook"

_genes, 2019, doi:10.3390/genes10110835_

Round 1

Reviewer 1 Report

Title: Transcriptomic revelation of phenolic compounds involved in Aluminum toxicity responses in roots of Cunninghamia lanceolata (Lamb.) Hook.

Dear Editor,

The current manuscript under consideration presents some interesting observation. The authors have done extensive work and presented some interesting observations.

However before it can be accepted for publication I have a minor concern: The resolution of figures: 1, 2, 3,6,12 and 14

Author Response

Thank you for pointing that the low-resolution of figure 1, 2, 3, 6, 12, 14. Indeed, the 300 dpi resolution of these figures maybe not meet our requirements. We have redraw these figures and some other low-resolution figures including figure 11 and figure 13, please see the changes in the attachment files with the name “genes-594517-_with track changes” and “genes-594517-_without track changes”.

We acknowledge your comments and suggestions very much, which are valuable in improving the quality of our manuscript.

Reviewer 2 Report

This manuscript presents a first transcriptome study of Cunninghamia lanceolata The data was generated and handled appropriately. I have a few minor edits/suggestions for the authors to address:

To get a sense of how completely the transcriptome assembly represents the gene space the authors should benchmark the transcriptome against a completeness index such as the BUSCO benchmark. In a few different sections the FDR cutoff is not correctly mentioned. Please fix this. Tables 1 & 2 belong in the supplementary material.  In section 3.4 when talking about KEGG pathways, the number of assigned DEGs should first be normalized to the number of assigned genes in each pathway. Figure 9 should show only TC2, since that is the Al toxicity response. TC1 DEGs can be shown in a supp figure. Figure 14 should compare the Fold change between Al1h vs. Al72h. This will make it easier to compare the RNA-Seq and qRT-PCR results.

Author Response

Point 1: To get a sense of how completely the transcriptome assembly represents the gene space the authors should benchmark the transcriptome against a completeness index such as the BUSCO benchmark. â€¨

 Response 1: Thank you for pointing these out. After removing adapters and low-quality reads, and filtering contaminant rRNA reads, clean reads were de novo assembled in the short read assembling program Trinity. BUSCO(Benchmarking Universal Single-Copy Orthologs) can provide quantitative measures for the assessment of genome assembly, gene set, and transcriptome completeness, based on evolutionarily-informed expectations of gene content from near-universal single-copy orthologs selected from OrthoDB. Although the genome of each species is different, there are always conservative sequences among closely related species. Based on this feature, BUSCO constructed single-copy sets of genes for several large evolutionary branches, according to the OrthoDB database. After obtaining the assembled genome or transcript sequence of a species, the assembled result can be compared with the conserved sequences in the gene set of the evolutionary branch to identify whether the assembled result contains these sequences, including single, multiple, partial or not. For transcripts, HMMER3 was used to compare reference gene sets after the longest ORF was identified.

Although we did not use BUSCO to evaluate the transcripts, we used it for evaluating the quality of genome assembly of Cunninghamia lancelanta, and we found that the ratio of the complete BUSCOs(c) is nearly 60% lower than empirical value 80%. However, if the complete BUSCOs value is lower than 80%, it is not present that some problems of assembly, and we should judge according to the actual situation, .

So, in this manuscript, we did not do BUSCO evaluation, maybe in the near future, we will combine genomic information and transcriptome for subsequent analysis, or perform transcriptome analysis with reference genome.  

Point 2: In a few different sections the FDR cutoff is not correctly mentioned. Please fix this. Tables 1 & 2 belong in the supplementary material.

Response 1: Thank you for pointing these out. We have modified the FDR in the manuscript, please see the changes in the attachment files with the name “genes-594517-_with track changes” and “genes-594517-_without track changes”.

Maybe the Table 1& 2 are not suitable in the text, we have moved them to the supplementary material, please see the modifications in the attachment files with the name “genes-594517-_with track changes” and “genes-594517-_without track changes”.

We acknowledge your comments and suggestions very much, which are valuable in improving the quality of our manuscript.

Point 3: In section 3.4 when talking about KEGG pathways, the number of assigned DEGs should first be normalized to the number of assigned genes in each pathway. Figure 9 should show only TC2, since that is the Al toxicity response. TC1 DEGs can be shown in a supp figure.

Response 1: Thank you for pointing these out. In section 3.4, KEGG pathway analysis were used the standard method, we have listed the number of candidate DEGs and All genes in each KEGG pathway, please see the attachment file with the name “Table S9 KEGG pathway of DEGs in TC1 and TC2 groups.”.

Pathway enrichment analysis identified significantly enriched metabolic pathways or signal transduction pathways in DEGs comparing with the whole genome backgraound. The calculating formula is:

Here N is the number of all genes that with KEGG annotation, n is the number of DEGs in N, M is the number of all genes annotated to specific pathways, and m is number of DEGs in M. The calculated p-value was gone through FDR correction, taking FDR≤0.05 as a threshold. Pathways meeting this condition were defined as significantly enriched pathways in DEGs.

In the beginning, we also separately displayed the DEGs in phenylpropanoid and flavonoid biosynthesis pathways between TC1 and TC2 groups, we found that difficult to compare the changes between two groups. Therefore, in order to show more clearly and quickly the variation of DEGs in phenylpropanoid and flavonoid biosynthesis pathways between TC1 and TC2 group, we chose to merge the figure of TC1(Red boxes outline squares representing TC1 DEGs) and TC2(squares lacking a red outline box represent TC2 DEGs).

We acknowledge your comments and suggestions very much, which are valuable in improving the quality of our manuscript.  

Point 4: Figure 14 should compare the Fold change between Al1h vs. Al72h. This will make it easier to compare the RNA-Seq and qRT-PCR results.

Response 1: Thank you for pointing these out. In the beginning, we have drawed heatmap by comparing the Fold change between Al-1h vs Al-72h, and we found that the results of RNA-Seq and qRT-PCR were all with the similar tendency, and this method only showed two columns in the heatmap. Of course, the above method could illustrate the accuracy of RNA-Seq. We also thought that if we used the data of two times and two treatment groups may more convincing to validate the accuracy of RNA-Seq, so we drawed the Figure 14 by using the data of Al1h: log2(Al-1 h/CK-1 h); Al72h: log2(Al-72 h/CK-72 h).

We acknowledge your comments and suggestions very much, which are valuable in improving the quality of our manuscript.
